# A Strong Link Between Oceanographic Conditions and Zooplankton δ^13^C and δ^15^N Values in the San Jorge Gulf, Argentina

**DOI:** 10.3390/biology13120990

**Published:** 2024-11-29

**Authors:** David Edgardo Galván, Manuela Funes, Flavio Emiliano Paparazzo, Virginia Alonso Roldán, Carla Derisio, Juan Pablo Pisoni, Brenda Temperoni, Daniela Alejandra del Valle, Valeria Segura, Seth D. Newsome

**Affiliations:** 1Centro para el Estudio de Sistemas Marinos (CESIMAR-CONICET), Boulevard Brown 2915, Puerto Madryn U9120ACD, Argentina; paparazzocnp@gmail.com (F.E.P.); pisonijp@gmail.com (J.P.P.); 2Instituto de Investigaciones Marinas y Costeras (IIMyC, UNMdP-CONICET), Juan B. Justo 2550, Mar del Plata B7608FBY, Argentina; manufunes15@gmail.com (M.F.); btemperoni@inidep.edu.ar (B.T.); 3Instituto Patagónico del Mar (IPAM-UNPSJB), Boulevard Brown 3051, Puerto Madryn U9120ACD, Argentina; 4Grupo de Investigación en Gestión Desarrollo Territorial y Ambiente (GesDTA-UTNFRCH), Facultad Regional Chubut, Universidad Tecnológica Nacional, Av. del Trabajo 1536, Puerto Madryn U9120QGQ, Argentina; virginia.a.roldan@gmail.com; 5Instituto Patagónico para el Estudio de los Ecosistemas Continentales (IPEEC-CONICET), Boulevard Brown 2915, Puerto Madryn U9120ACD, Argentina; 6Instituto Nacional de Investigación y Desarrollo Pesquero, Paseo Victoria Ocampo Nº1, Mar del Plata B7602HSA, Argentina; cderisio@inidep.edu.ar (C.D.); ddelvalle@inidep.edu.ar (D.A.d.V.); vsegura@inidep.edu.ar (V.S.); 7Consejo Nacional de Investigaciones Científicas y Técnicas (CONICET), Buenos Aires C1425FQB, Argentina; 8Biology Department, University of New Mexico, Albuquerque, NM 87131-0001, USA; newsome@unm.edu

**Keywords:** isoscapes, stable isotopes, frontal systems, southwest Atlantic Ocean, Patagonia

## Abstract

Stable isotope analysis provides valuable data for marine ecology research. For proper interpretation of results, baseline data from organisms at or near the base of the food web is critical. This study analyzes how stable isotope ratios in small zooplankton (copepods) vary across San Jorge Gulf and relates the observed variation to physical and chemical oceanographic conditions. San Jorge Gulf is a basin of 40,000 km^2^ located in the Argentine Sea (southwest Atlantic Ocean). The area is of great importance, supporting several natural contributions to people, such as large fisheries and ecotourism based on the marine mammal and seabird populations. We sampled copepods, as they feed on phytoplankton and are robust tracers of processes occurring at the base of the food web. The results revealed significant variation in both carbon and nitrogen stable isotope ratios. This variation was related to nutrient concentration, water column depth, and oceanographic processes that mix waters, all factors impacting phytoplankton growth rates. Our findings provide baseline data to address questions related to the feeding and movements of other animals inhabiting the region.

## 1. Introduction

Carbon (δ^13^C) and nitrogen (δ^15^N) isotope analysis is a powerful and routinely used tool in marine trophic ecology [1,2]. Baseline δ^13^C and δ^15^N values of organisms at or near the base of the marine food webs are critical for accurate interpretation of the trophic dynamics of organisms [3], the food web structure [4], and the relative reliance on benthic versus pelagic sources of energy [5]. Baseline data can be constructed from the δ^13^C and δ^15^N values of primary producers [6] or primary consumers [7], but their isotopic composition varies over time and space due to physicochemical and biological processes. This spatial variation is represented by isoscapes, which are maps of interpolated isotope values collected from a limited set of locations, spanning areas that vary in extent from tens [8] to thousands [9] of square kilometers. Understanding how δ^13^C and δ^15^N values are predictably influenced by oceanographic variables provides insights into ecosystem functioning [7], crucial for interpreting consumer-derived isoscapes [10], enabling the creation of dynamic isoscapes [11] that represent isotopic variation in both time and space.

In marine ecosystems, baseline isoscapes are usually derived from sessile or low-motility primary consumers, such as bivalves [5,12] or zooplankton [5,7], under the assumption that they effectively reflect and integrate the isotopic variation of local primary producers. In marine ecosystems, various physicochemical and biological factors influence the δ^13^C and δ^15^N values of phytoplankton. The extent of δ^13^C discrimination during photosynthesis is primarily influenced by: (1) the concentration and isotopic composition of dissolved inorganic carbon in seawater, (2) phytoplankton growth rates [13], and (3) cell size [14,15], which is often related to assemblage composition [6]. Larger phytoplankton cells are ^13^C-enriched compared to smaller cells due to passive CO_2_ diffusion into the cell [16]. Furthermore, there is a negative relationship between the external CO_2_ concentration and phytoplankton growth rate, such that a decrease in CO_2_ concentration reduces ^13^C discrimination [6]. In temperate systems, this relationship is observed during seasonal blooms when high growth rates result in increased δ^13^C values of phytoplankton due to reduced discrimination between CO_2_ and fixed organic carbon [13].

Phytoplankton growth rate and community composition are primarily controlled by the concentrations of three macronutrients: nitrogen (N), phosphorous (P), and silica (Si) [17]. Measuring the availability of N is crucial for interpreting isoscapes, and there are four forms of inorganic nitrogen in ocean systems: NO_3_^−^, NO_2_^−^, NH_4_^+^, and N_2_. NO_3_^−^ (nitrate) has the fastest assimilation rate, and as such is the primary form fueling primary production in most areas of oceans [18]. In most regions, N is limiting, so the average isotopic composition of phytoplankton mirrors that of nitrate [19]. The concentration of the other macronutrients, Si and P, could also influence δ^13^C and δ^15^N values at the base of pelagic marine food webs. Under certain conditions, primary producers can deplete available nutrients, reducing their concentrations to levels that limit growth rates and influence δ^13^C and δ^15^N discrimination. For example, in the Argentine outer shelf, the Si:N ratio positively correlates with diatom abundance, which influences their δ^13^C values [20].

There is scarce spatially explicit baseline information on δ^13^C and δ^15^N values on the Argentine shelf in the western South Atlantic Ocean (~35° S–56° S). In the middle and outer shelf (100 m–200 m deep), δ^13^C values of surface particulate organic matter (POM) showed a strong correlation with sea surface temperature (SSTs) [20], which likely reflects temperature-related variations in aqueous CO_2_ concentrations and their effect on carbon isotopic discrimination by marine phytoplankton [20]. These patterns appear to be passed up regional food webs, as whole-blood δ^13^C values of Magellanic penguins (*Pheniscus magellanicus*) decrease with increasing latitude of collection [21]. Regarding δ^15^N values, SST and nutrient pools including unutilized nitrate, Si:N ratios, and phosphate concentration were identified as key drivers of POM δ^15^N values on the middle and outer Argentine shelf [20]. In coastal waters between 41° S and 53° S, δ^15^N isoscapes of penguin whole blood revealed two distinct zones: lower δ^15^N values south of ~47° S and higher values to the north. The transition between these zones occurs within the San Jorge Gulf (SJG), a region of particular importance.

The SJG is a semicircular basin (~40,000 km^2^) located on the Argentine shelf (Figure 1), one of the most productive continental shelves in the world supporting large and lucrative invertebrate and vertebrate industrial fisheries [22,23]. Due to its unique oceanographic and ecological features, as well as its susceptibility to human impacts, the SJG has been designated as one of the five priority research areas by the Argentine government under the Pampa Azul initiative (http://www.pampazul.gob.ar accessed on 31 October 2024). The SJG’s waters are sourced from the south, primarily via the Patagonian Current, which is influenced by cold and low-salinity waters from the Magellan Strait [24,25]. Circulation within the SJG varies seasonally and is regulated by atmospheric heat fluxes, which generate a cyclonic gyre for most of the year [26,27]. The SJG has no significant freshwater inputs, as it is located in a semi-desert region that receives ~250 mm rain year^−1^ and lacks river discharges.

The bioavailability of macronutrients (especially nitrate) for primary producers in the SJG is primarily driven by thermal stratification of the water column [28] and its disruption in frontal areas. Frontal areas within the gulf are generated by the interaction of tidal currents with bottom topography [29,30], islands, and headlands [31] or driven directly by winds [32]. Two frontal systems, referred to as the Northern Patagonian Frontal System (NPFS) and the Southern Patagonian Frontal System (SPFS), are the areas of highest productivity (Figure 1) [33,34]. The NPFS develops during austral spring and summer along the inner shelf and extends into the northern half of the SJG (Appendix A). This frontal system is driven by seasonal thermal stratification and high dissipation of tidal energy [33,35]. In contrast, the SPFS is a permanent thermohaline front characterized by the transition between the tidally mixed, nutrient-rich, and seasonally stratified waters of the Patagonian Current and the relatively saline and seasonally stratified inner waters [25,33].

**Figure 1 biology-13-00990-f001:**
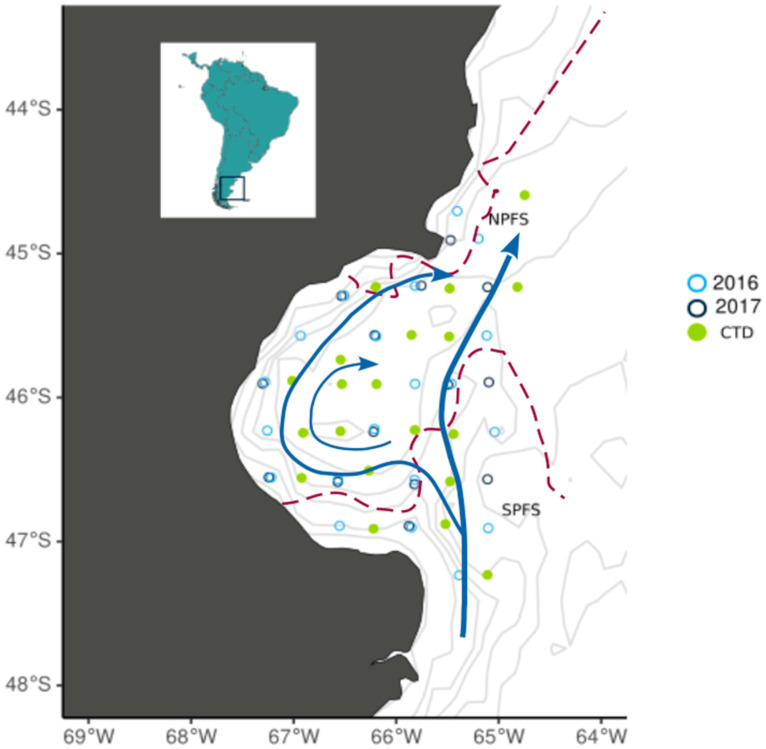
Map showing grid sampling design; dark blue and light blue circles denote zooplankton and water sampling locations and green circles denote stations where only oceanographic conditions were measured via CTD. The average position of the Northern Patagonia Frontal System (NPFS) and the Southern Patagonia Frontal System (SPFS) were sourced from [33] and are shown as red dashed lines. The main current direction was sourced from [27] is shown as blue arrows.

The diverse physical and chemical conditions of the SJG promote high biological production [36,37,38], with spatio-temporal variation in phytoplankton [39] and zooplankton [40,41] abundance. Spring phytoplankton blooms strongly associated with the two frontal systems are dominated by diatoms and dinoflagellates [36,37,38]. The magnitude of spring primary production is comparable to that reported for the rich Argentinian shelf break [38] and supports a productive local zooplankton community dominated by copepods that account for >80% of the biomass in the zooplankton community. Medium and large calanoid copepods (>1 mm total length) are primarily herbivorous/omnivorous [41], whereas relatively small (<1 mm) cyclopoid copepods exhibit more omnivorous and/or detritivorous foraging behavior [42,43]. In addition to these trophic preferences that likely influence the isotopic composition of copepods, the δ^13^C and δ^15^N values of zooplankton increase within the SJG from north to south during the austral summer in response to environmental physico-chemical variation [40], specifically SST and inorganic N availability [44]. A similar latitudinal pattern has been reported for the outer Argentine shelf [20]. These spatial patterns also agree with the δ^13^C and δ^15^N isoscapes constructed from penguin tissues [21]. Additionally, dust inputs derived from westerly windstorms [45] may play an important role in regulating baseline isotope values in this region.

To enhance our understanding of how physical and chemical oceanographic conditions influence stable isotope values at the base of the food web, we analyzed spatial and interannual variation in δ^13^C and δ^15^N values of zooplankton collected in the SJG for comparison to local physical and chemical oceanographic data, including water depth, surface/bottom water temperature, and macronutrient (N, P, and Si) concentrations. This study was conducted during the austral spring, when primary production is expected to be at its highest [34]. We chose to target zooplankton sized between 200 and 300 μm (prosome width), because this assemblage primarily consists of immature copepods, assumed to be primary consumers [5,40], and as such are robust tracers of short-term spatial variation in δ^13^C and δ^15^N values [7,10].

## 2. Material and Methods

Samples and data were collected during the austral spring, from 12 to 26 November 2016 and from 28 October to 8 November 2017, aboard the R/V *Puerto Deseado* (CONICET, Buenos Aires, Argentina). The sampling comprised a regular grid (Figure 1 and Appendix A). At each sampling station, profiles of temperature, conductivity (salinity), and pressure were measured using a CTD SBE 911 Plus©, Sea-Bird Scientific, Washington, EEUU. Water column stability was estimated using the Brunt–Väisälä frequency [46] calculated from potential density values integrated in the water column up to the maximum depth reached at each station (<100 m). Seawater samples were collected using a rosette system equipped with Niskin bottles near the surface (5 m depth, hereafter referred to as ‘surface’) and bottom (5 m above the seafloor, hereafter referred to as ‘bottom’). Samples were collected to measure nitrate, phosphate, and silicic acid concentrations (Figure 1). These were preserved at −20 °C in polyethylene bottles previously washed with distilled water and 5% HCl and then rinsed with distilled water. Once on land, macronutrient concentrations were determined using a Skalar San Plus autoanalyzer (V.B. 2005a,b,c), Skalar Analytical^®^, Breda, The Netherlands.

Zooplankton samples were collected at 21 stations in 2016 (Figure 1) and 14 stations in 2017 by oblique tows with a *Minibongo* net (200 µm mesh). Fewer stations were sampled in 2017 due to the time constraints of the cruise. Both the 2016 and 2017 samplings were conducted on a regular grid covering similar areas (Figure 1 and Appendix A), with 14 stations shared between the two years. The primary difference between the samplings is that, in 2016, two additional stations were located outside the SJG—one to the north and one to the south—while five were interspersed within the gulf, maintaining the same boundary at the gulf’s mouth. After each tow, we separated the fraction of zooplankton sized between 200 and 300 μm of prosome width, which mainly included immature copepods between 1 and 1.5 mm in size (total length), by sieving the samples through a 300 µm mesh. Then, samples were stored at −20 °C until processing for stable isotope analysis. This size fraction has been previously used to estimate pelagic isotopic baselines by [5] in the SJG. Zooplankton samples were dried at 60 °C for 72 h and homogenized into a fine powder, and δ^13^C and δ^15^N values were measured via a Costech 4010 elemental analyzer coupled to a Thermo Scientific Delta V Plus isotope ratio mass spectrometer at the University of New Mexico Center for Stable Isotopes (Albuquerque, NM, USA). Internal standards used were soy, tuna, whey protein, casein, IAEA N1, IAEA N2, USGS 42, and USGS 43. Mean within-run standard deviation (SD) of a suite of in-house (soy, tuna, casein) and internationally accepted (IAEA N1, IAEA N2, USGS 42 and USGS 43) reference materials were 0.1‰ for both δ^13^C and δ^15^N values. Given that samples with C/N ratios greater than 3.5 were observed, and that lipid extraction was not performed prior to stable isotope analysis, δ^13^C values were lipid-normalized using the general equation proposed for aquatic organisms [47]. All subsequent data analyses and results were conducted using lipid-normalized δ^13^C values.

To determine the natural abundance of ^13^C (δ^13^C) of particulate organic carbon (POC) at eight selected stations (four in the south and four in the north of the SJG each year), surface seawater samples were collected in polycarbonate bottles and immediately filtered onto pre-combusted Whatman GF/F glass-fiber filters. The filters were stored dry, and once on land they were fumed with HCl and encapsulated. The δ^13^C of POC in the samples was analyzed using an Elementar Vario Micro Cube elemental analyzer (Elementar Analysensysteme GmbH, Hanau, Germany) interfaced with a PDZ Europa 20-20 isotope ratio mass spectrometer (Sercon Ltd., Cheshire, UK) at the Stable Isotope Facility of the University of California in Davis (USA).

A *Bongo* net (300 µm mesh) equipped with flowmeters was used to estimate zooplankton community composition at a subset of stations. *Bongo* net samples were fixed immediately after collection with a 5% formalin–seawater solution. They were inspected in the laboratory under a Wild M5 stereoscopic microscope to estimate both the abundance and the taxonomic composition of the 1 mm–2 mm zooplankton fraction, classifying the species according to their trophic function *sensu* [41,48,49].

We analyzed the relationship among physical and chemical oceanographic variables and zooplankton δ^13^C and δ^15^N values using generalized linear mixed models (GLMMs), assuming a Gaussian error distribution. The GLMM aimed to investigate the temporal and spatial variation of these variables. We included water column depth, sea surface temperature (5 m depth), near-bottom temperature (5 m above the bottom), and water column stability. Nitrate, phosphate, and silicic acid concentrations at the surface and bottom were included as predictors of nutrient availability. Station ID and year were treated as random variables. Before conducting the GLMM, we performed a data exploration process following a standard procedure [50]. This process included outlier detection, assessment of variance heterogeneity, checking for collinearity among predictors, and selection of predictors using a variance inflation factor (VIF) analysis. Initially, complete models were fitted, incorporating all non-collinear predictors, and alternative structures for the random factor were compared. Subsequently, based on the best random structure, a backward elimination procedure was employed to select the most parsimonious model and identify the best predictors. The criteria for the backward elimination procedure were the lowest Akaike’s Information Criterion (AIC) values and a delta AIC > 2 [51]. To examine spatial autocorrelation among residuals, we applied Moran’s I coefficient, as described in [52]. Furthermore, we evaluated goodness of fit and deviations from model assumptions by the Shapiro–Wilk test of normality, fitting linear models using the residuals and absolute values of the residuals against the predicted values and visually inspecting a QQ plot, which compares the quantiles of the theoretical residuals under the assumption of normality to the quantiles of the observed residuals [51]. To facilitate interpretation of the results, we identified the stations where macronutrients might have limited primary producers according to the thresholds proposed by [53]: nitrate < 0.7 μM and silicic acid < 1.8 μM. The statistical software program R [54] was used for data analysis, with the nlme package [55] for model fitting and the ape [56], gstat [57], and sp [58] packages for assessing spatial autocorrelation. In addition, the correlation between zooplankton and phytoplankton δ^13^C values was examined using linear regression and compared to an assumed slope of 1 and an intercept of 1‰ (a 1:1 relationship with a discrimination of 1‰). The regression model was fitted using mean δ^13^C values of zooplankton for the 12—six stations per year—samples of phytoplankton. Model assumptions were tested using the same procedure previously described.

## 3. Results

Oceanographic conditions differed between the two years. The stability of the water column suggests a larger area of stable waters with low vertical mixing and well-established frontal systems in 2016, particularly the SPFS, compared to 2017 (Figure 2 and Table 1). Additionally, the range between the extreme values of surface and bottom temperatures was broader in 2016 compared to 2017 (Table 1).

Zooplankton δ^13^C and δ^15^N values varied among sampling stations, with 2016 showing greater variability compared to 2017 (Table 1 and Figure 3). δ^15^N values generally increased from south to north and from offshore to inshore waters in both years (Figure 3), a pattern that was more pronounced in 2016 than in 2017. Lower δ^13^C values were observed in the central region of the SJG, and likewise this pattern was more pronounced in 2016 than in 2017 (Figure 3).

The exploratory data analysis revealed correlations between macronutrient concentrations and water column structure (Figure 4), except for surface silicic acid. Macronutrient concentrations near the bottom exhibited correlations among each other and increased with depth (Figure 4). Nitrate and phosphate at the surface were also positively correlated. However, surface silicic acid did not show a strong correlation with any of the studied variables. In agreement with the results of the correlation matrix, the variables with the most information (VIF < 2) selected as predictors for the GLMMs were: water column depth, water column stability, and surface nitrate and silicic acid concentrations.

Data analysis by GLMM showed that year and sampling station had intrinsic differences, as the best structure for the random components included a variable intercept for the stations nested in the years (Appendix A). In the final simplified model, δ^15^N values were strongly and negatively related to surface nitrate (Table 2). This relationship was evident in the spatial patterns in 2016 (Figure 3 and Figure 5) and was fully captured in the model. Surface silicic acid and water column stability were also significant variables in the model (Table 2). Higher surface silicic acid concentrations, and more stable water columns were related to ^15^N-rich zooplankton (Figure 2 and Figure 6). The model had a good fit, as residuals did not show spatial autocorrelation (I = 1.1 × 10^−4^
*p* = 0.73) or departure from the assumptions of normal distribution of errors (W = 0.99, *p* = 0.85) and homogeneity of variance (Appendix A). For δ^13^C values, the selected model showed a negative correlation with water column stability and surface nitrate concentrations (Table 2). A relationship between stable waters with low vertical mixing at the center of the SJG and low zooplankton δ^13^C values was particularly evident in 2016 (Figure 2 and Figure 3). Additionally, water column depth significantly contributed to the model, with zooplankton in coastal waters exhibiting lighter isotopic values than those in central and outer waters (Table 2 and Figure 3). The residuals from the model did not show signs of spatial autocorrelation (I = −2.5 × 10^−3^, *p* = 0.84) or departure from the assumptions of hnormal distribution (W = 0.99, *p* = 0.51) and homogeneity of variance (Appendix A).

The surface nitrate concentration was highest in the southern Gulf, with values close to 11 µM (Table 2, Figure 5). As circulation progressed in the SJG northward, it decreased, likely due to stratification and phytoplankton consumption. Consequently, in 2016, numerous northern stations were observed with values below the minimum concentration required for the normal development of primary producers (<0.7 µM) (Figure 3). The other macronutrient with a significant effect, silicic acid, followed a similar trend to nitrate (Figure 6). Additionally, the selective consumption of this macronutrient led to limiting concentrations in certain areas for the organisms that require it for their development (<1.8 µM) (Figure 3). For both macronutrients, this pattern was more pronounced in 2016 than in 2017.

Zooplankton δ^13^C values correlated well with phytoplankton δ^13^C values (r^2^ 0.60, Figure 7). The regression model exhibited a slope of 0.75 (sd = 0.21, *p* = 0.005), which included the theoretical 1:1 relationship with a discrimination of 1‰ within its 95% confidence interval. Although there are two extreme points of high leverage, the result adds support to the assumption of a predominance of zooplankton feeding on phytoplankton.

The taxonomic description of zooplankton samples showed that the fraction of zooplankton smaller than 2 mm comprised immature copepods (copepodites) and some adult copepods (Table 3). Copepod abundance was higher in 2016 than 2017 (Figure 8). In 2016, copepod abundances varied between 42 and 2571 ind m^−3^ (mean = 554; SD = 641), with the highest abundances found in the northern sector of the SJG. In 2017, abundances varied between 63 and 501 ind m^−3^ (mean = 222; SD = 120), but the highest abundances occurred towards the southern sector and inner waters of the SJG. Copepodites presented similar abundances in both years, but 2017 had less variability among stations than 2016 (Figure 8). Copepod community composition was similar in both years. The most abundant species, in decreasing order, were *Ctenocalanus vanus*, *Drepanopus forcipatus*, and *Acartia tonsa*, while *Oithona aff. helgolandica* and *O. atlantica* were less abundant. Copepod assemblages showed similar spatial patterns in specific and functional composition in both years (Figure 8). The herbivore *D. forcipatus* was only present in the southern sector, while the central and northern sectors were dominated by herbivore-omnivores (*C. vanus*) and omnivore-detritivores (*A. tonsa, O. aff. helgolandica*, and *O. atlantica*).

## 4. Discussion

Our results show that copepod δ^13^C and δ^15^N values varied by up to 7–8‰ over a relatively small spatial scale (200–300 km) across the SJG. Analysis of zooplankton assemblage composition indicated that herbivorous copepods dominated both years. In addition, the observed relationship between POM and zooplankton δ^13^C values suggests that zooplankton isotopic composition was a good proxy for that of phytoplankton at the base of the food web. We identified distinct spatial patterns for δ^13^C and δ^15^N values, including lower δ^13^C values in the center of the SJG that were inversely related with water column stability, surface nitrate concentration, and water column depth. In contrast, δ^15^N values generally increased from south to north, showing an inverse relationship with surface nitrate bioavailability and water column stability and a positive relationship with surface silicic acid concentration. Below we explore the potential abiotic and biotic factors that influence the observed spatial patterns in copepod isotopic variation, with special attention to the bioavailability of nitrate in the photic zone as influenced by the dominant northward current and seasonal development of frontal systems in the SJG.

The oceanographic conditions in SJG are likely to exhibit substantial interannual variability [34], with even sporadic upwelling events having a great effect on primary productivity [44]. Our results show that spring water column stability was similar in both years, although it exhibited a broader range in 2016. Reduced stability was observed in the south–southwest region during 2016, whereas stability in the northeast was lower in 2017. Similar findings were found using the Simpson stability parameter for the 2016 and 2017 cruises [59], which were lower than the values estimated for the spring of 2008 [38] and the spring of 2019 [59]. These observed differences among years may also depend on the timing of data collection and the temporal progression of frontal development. Spring sampling started ~15 days earlier in 2017 than in 2016, which may partially explain observed differences between years. Furthermore, the spring 2008 surveys reported in [38,59] were conducted in late November and early December, when the water column is expected to be more stable, whereas the published data for spring 2019 [59] were collected at a similar time as the spring 2016 and 2017 data reported here. These differences in water column stability between years, together with the temporal process of frontal development, are likely influencing the isotopic composition of primary producers and consumers at the base of the food web.

δ^13^C values in marine ecosystems generally decrease with increasing latitude [60,61] due to the temperature-related effects of CO_2_ solubility on carbon isotope discrimination between aqueous CO_2_ and fixed organic carbon [20]. In highly productive areas, however, dissolved CO_2_ concentrations are also controlled by phytoplankton growth rates [6]. We did not observe a positive correlation between latitude and copepod δ^13^C values, likely because spatial patterns in productivity and water column structure do not follow a latitudinal pattern across the SJG. The SJG is located at temperate latitudes (between 45° S–47° S) where the surface heat flux tends to stratify the water column during the spring, which dampens the supply of nutrients from bottom waters. However, the complex topography near the mouth of the southern sector of the SJG, in conjunction with tidal currents, homogenizes the water column [29,62], bringing cold, nutrient-rich waters from the Magellan Plume to the surface. Moreover, at the mouth of the SJG in the southern frontal zone, tides facilitate the influx of low-salinity, nitrate-rich waters, which provides a lateral fertilization mechanism to maintain the surface chlorophyll layer [63]. In addition to these mechanisms, intense winds in coastal areas of the southwestern sector of the SJG can lead to upwelling that mixes the water column [26,32,44]. This local process was evident in 2016, when a strong windstorm caused coastal upwelling at 46.5° S, destabilizing water column stratification and locally increasing surface nitrate concentrations [32,44]. In the northern sector near the coast, tidal currents and topography interact to vertically mix the water column [64,65], generating a highly turbulent region [31] that in the warm (productive) season is fed by nutrient-rich waters originating from the center of the SJG [30]. These conditions combine to create areas with cold bottom waters upwelled to the surface that enhance nutrient availability in the southern and northern sectors of the SJG, which by extension increases local phytoplankton growth rates [28,44]. Such factors affect carbon isotope discrimination, leading to an increase in phytoplankton δ^13^C values.

The spatial pattern in δ^13^C values described above was more evident in 2016 than in 2017 (Figure 2). This difference is likely because of differences in the time of sampling between years. In 2017, sampling occurred just after the formation of a weak seasonal frontal system [59], which resulted in lower primary production (phytoplankton biomass) in comparison to the previous year [59]. As zooplankton quickly integrate changes in the isotopic composition at the base of the food web through grazing and assimilation of phytoplankton, environmental conditions along with phytoplankton composition and growth rates in the days prior to sampling are crucial for accurate data interpretation. In 2016, chlorophyll *a* concentrations estimated from satellite-derived color imagery that serves as a proxy of phytoplankton biomass were higher in the northern and southern regions of the SJG compared to the central area during the week leading up to sampling (Appendix A). However, there was no equivalent increase in chlorophyll *a* concentrations in the frontal areas before the cruise in 2017, which was conducted two weeks earlier than in 2016 (Appendix A). This short delay meant that, in 2017, phytoplankton had less time to capitalize on favorable spring conditions, such as increased light availability and abundance of nutrients. Overall, this resulted in higher zooplankton δ^13^C values in 2016 relative to 2017 (Figure 3), which is a product of the observed interannual differences in primary productivity in the frontal area at the time of sampling between these two years (Figure 6).

A previous study [40] sampled the zooplankton community of the SJG during the summer of 2014 and reported a range of mean δ^13^C values (−25‰ to −18‰) similar to what we observed, but with a general increase from north to south. During the summer of 2014, the northern SJG was stratified, while the southern sector exhibited a homogeneous water column [40]. Thus, prior results are consistent with the patterns reported here, showing that zooplankton δ^13^C values are inversely correlated with the vertical structure of the water column, being lower in the region where conditions favor nutrient input from deep waters.

Another factor explaining the observed δ^13^C values was water column depth, as the SJG deepens toward its center. Specifically, δ^13^C values were inversely correlated with water column depth, being lower in the center of the SJG and showing the expected trend from coastal to deeper zones [7]. However, the relationship between water column depth and δ^13^C values is likely indirect and mediated by water column stability. Both water column depth and stability had a weak positive correlation (Figure 4), as deeper waters tend to have more stable vertical profiles.

On the outer Argentine shelf, POM δ^15^N values increase from ~2.5‰ at 44° S to 5.6‰ at 54° S [20]. Our zooplankton results show a similar latitudinal pattern, which was more pronounced in 2016 than in 2017 (Figure 3). Nitrate and silicic acid might have limited phytoplankton growth in much of the study area during 2016–2017 (Figure 3). A negative correlation between POM δ^15^N values and nitrate and phosphate concentrations was reported for the outer Argentine shelf, suggesting that nitrate availability was likely the main driver of latitudinal variation in δ^15^N [20]. Our results for the SJG agree with this mechanism, as zooplankton δ^15^N values and surface nitrate concentrations were negatively related. In addition, we found that surface nitrate and phosphate concentrations were highly correlated (Table 2). In agreement, there is a previous description of a northward increase in zooplankton δ^15^N values for the zooplankton community within a range comparable with our results (6–15‰) [40]. Following the same line of evidence and reasoning previously offered to describe patterns in δ^13^C values, phytoplankton may uptake ^15^N-depleted nitrate from the Magellan Plume in the southern sector of the SJG, which is available at the surface along frontal zones [66]. Consequently, the nitrate concentration is likely to be reduced by biological uptake, leading to low concentrations at the center of the SJG (Figure 5) and concentrations below the limiting concentration in the northern sector of the gulf [44]. The bottom waters in the center of the SJG are rich in organic matter that supports a community of microbial decomposers and benthic scavengers (e.g., *Pseudechinus magellanicus* and *Grimothea gregaria*; [67]) that generate ^15^N-enriched inorganic nutrients [28] that are upwelled to the surface by the northward circulation pattern [30] during the warm season in the northern frontal zone. Combined, ^15^N-depleted nitrate in the south and ^15^N-enriched ammonium in the north [44] support primary production in the SJG to create the observed latitudinal pattern in zooplankton δ^15^N.

In addition to the process described above, copepod δ^15^N values may have been influenced by a spatial shift in trophic level related to resource availability. In this scenario, δ^15^N values would be expected to increase if omnivory becomes a more prevalent foraging strategy. Estimates of copepod trophic position were generated from adults (up to 2 mm in size) collected with a 300 µm mesh net, which is larger than the fraction used for stable isotope analysis that consisted of slightly smaller and immature copepods in the CIV–CV copepodite stages (1–1.5 mm). These slightly smaller copepods likely had similar dietary preferences to the adults, as previously described for calanoid copepods [68,69]. Herbivorous species dominated the southern region of the gulf, while the northern region consisted of a combination of herbivore-omnivores and omnivore-detritivores. This distribution in the functionality of the copepod community aligns with expectations for spring and summer, when the most abundant species in the size fraction used for stable isotope analysis were *C. vanus*, followed by *D. forcipatus* and *A. tonsa* [40,41]. Both *C. vanus* and *A. tonsa* are typically herbivores when phytoplankton is abundant; however, they may exhibit omnivorous behavior and also utilize heterotrophic microorganisms [48] when phytoplankton is less available. Contrary to the expected trend of a trophic level increase, our findings indicate that the highest δ^15^N values observed coincided with high phytoplankton availability; e.g., in the northern SJG in 2016 [42]. While the influence of omnivory cannot be completely ruled out, we suggest that the primary drivers of zooplankton δ^15^N are spatial gradients in the isotopic composition of inorganic nitrogen sources available for primary production.

## 5. Conclusions

Our results represent the first reconstruction of the spatial variation in zooplankton δ^13^C and δ^15^N values across the SJG, with interpretations based on a combination of physical and chemical oceanographic data. This is a crucial step in developing dynamic isoscapes for this productive region that supports high biodiversity; however, it is important to acknowledge the limitations of our study. Our inferences are based on two sampling periods conducted during the austral spring when frontal zones were established, especially in 2016. Since the SJG is in a temperate region, it will be essential to conduct year-round sampling to gain a better understanding of the geographic patterns and the influence of various oceanographic variables on shaping the δ^13^C and δ^15^N values of primary producers and consumers in this region. Additionally, to develop baseline data representative of the different energy pathways that fuel this productive and diverse continental shelf food web, it would be beneficial to sample other basal pelagic (e.g., euphausiids) and benthic (e.g., filter feeders) consumers in future research efforts.

## Figures and Tables

**Figure 2 biology-13-00990-f002:**
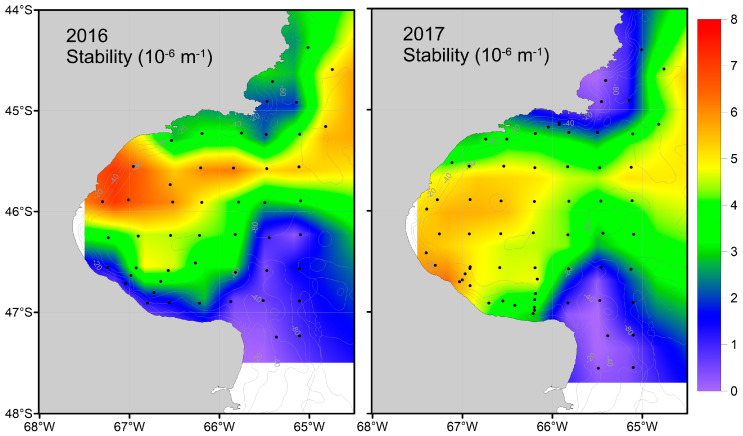
Water column stability during 2016 (**left**) and 2017 (**right**). Low values correspond to strong vertical mixing, while high values correspond to stable waters with low vertical mixing.

**Figure 3 biology-13-00990-f003:**
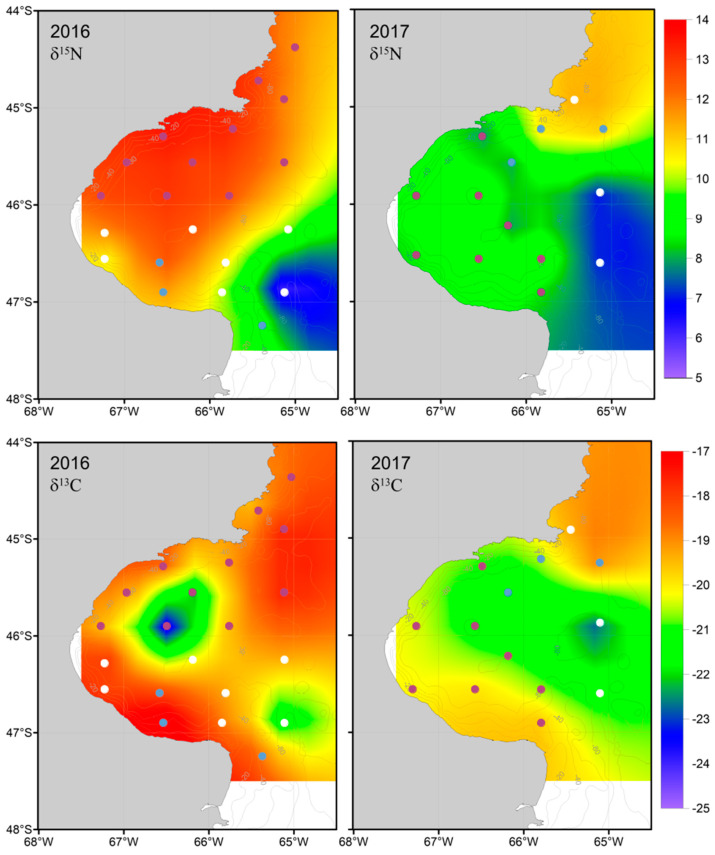
Maps of zooplankton δ^15^N (**top**) and δ^13^C (**bottom**) values for 2016 (**left**) and 2017 (**right**). Circles show the station locations, with colors indicating sites where macronutrients with significant effects in the GLMMs may have limited primary production. White: no limitant; purple: surface nitrate and silicic acid limitant; light blue: surface silicic acid limitant.

**Figure 4 biology-13-00990-f004:**
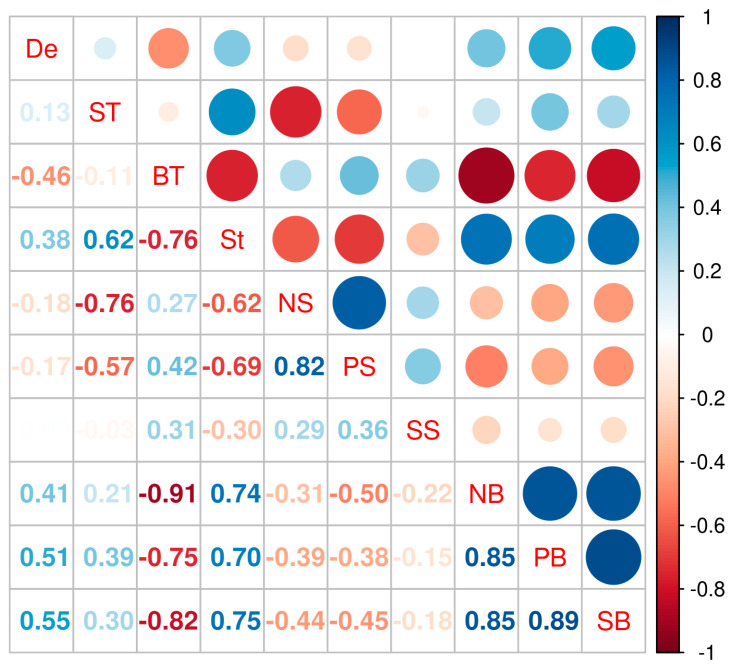
Pearson correlation among oceanographic variables. Water column depth (De in m), temperature at the surface (TS in °C), temperature near the bottom (TB in °C), water column stability (St in m^−1^), and nitrate, phosphate, and silicic acid concentrations at the surface (NS, FS, and SS in μM) and near the bottom (NB, FB, and SB in μM). Correlation coefficients are expressed numerically in the lower left quadrant and represented graphically as colored bubbles in the upper right quadrant.

**Figure 5 biology-13-00990-f005:**
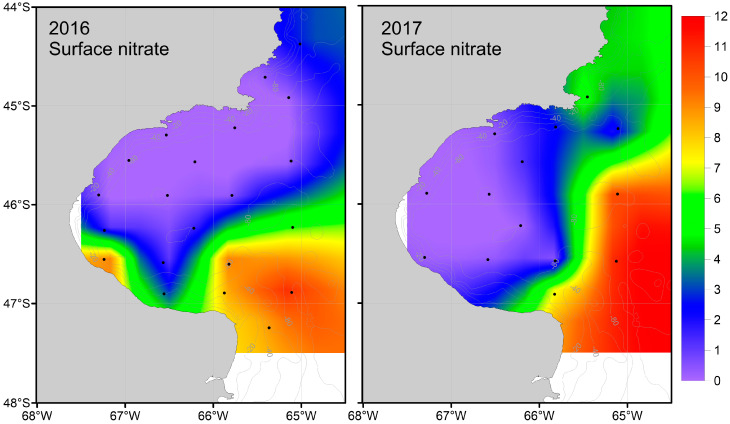
Maps of surface water nitrate (NO_3_^−^) concentration in 2016 (**left**) and 2017 (**right**).

**Figure 6 biology-13-00990-f006:**
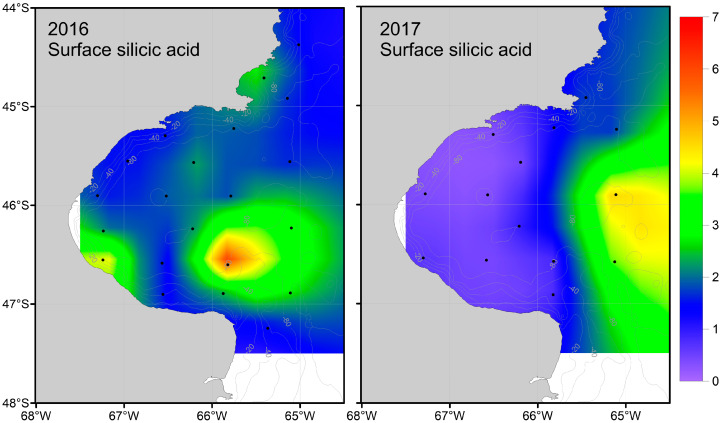
Maps of surface water silicic acid concentration in 2016 (**left**) and 2017 (**right**).

**Figure 7 biology-13-00990-f007:**
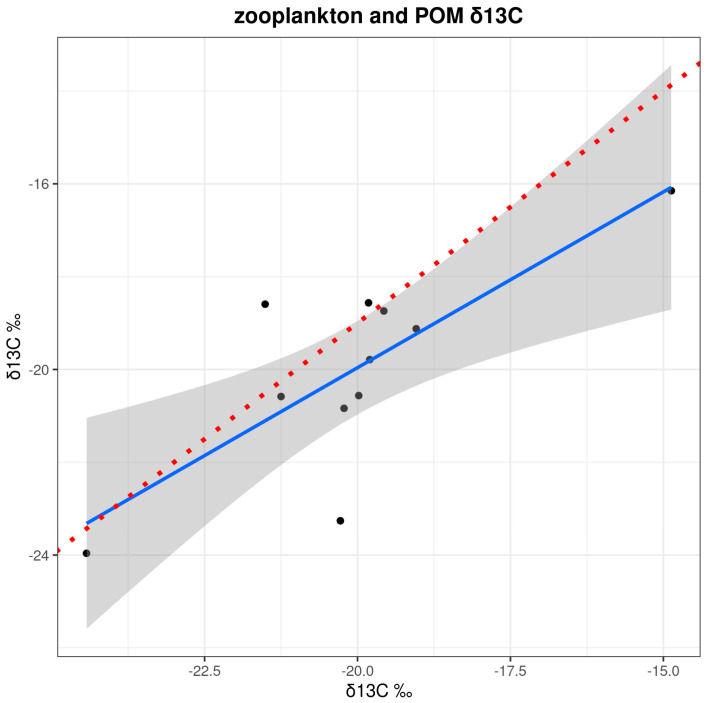
Linear regression between zooplankton and POM δ^13^C values (r^2^ = 0.60). The blue line is the observed relationship (b = 0.75, a = −4.8, *p* = 0.005), the shaded area indicates the 95% confidence interval, and the red line represents a theoretical 1:1 relationship with a discrimination of 1‰, assuming herbivory.

**Figure 8 biology-13-00990-f008:**
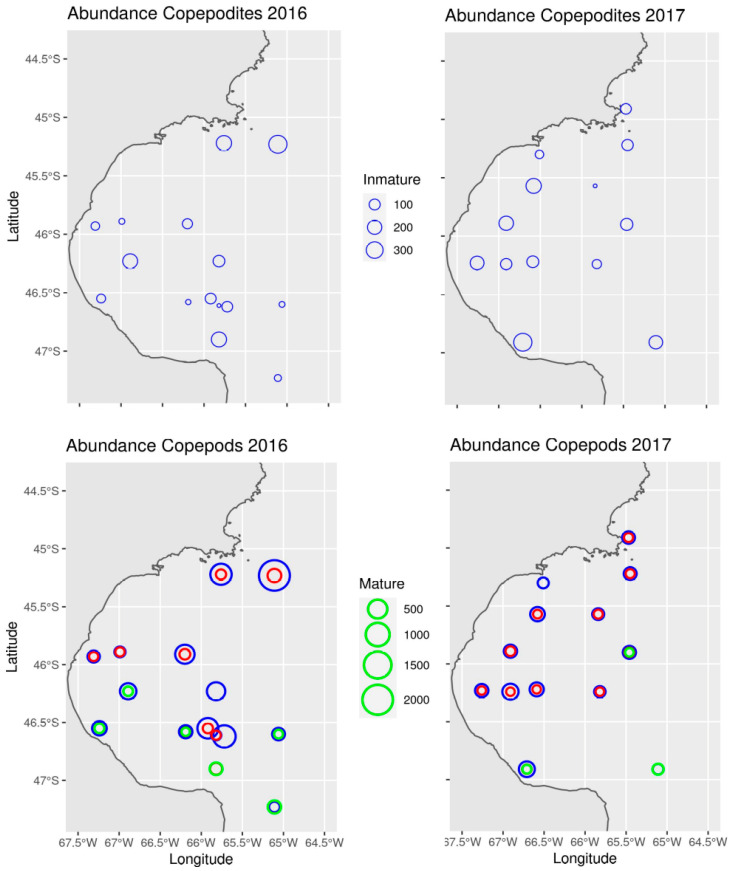
Abundance maps (ind m^−3^) of immature (**top**) and adult (**bottom**) copepods in 2016 (**left**) and 2017 (**right**). Adult copepods are grouped by trophic function: herbivores (green), herbivore-omnivores (blue), and omnivore-detritivores (red).

**Table 1 biology-13-00990-t001:** Summary of measured oceanographic variables and zooplankton isotopic values for the years 2016 and 2017.

	2016			2017		
	Mean	SD	Range	Mean	SD	Range
Surface Temperature (°C)	11.4	0.9	9.4 to 12.5	11.0	0.6	9.9 to 11.9
Bottom Temperature (°C)	9.9	0.9	8.4 to 11.6	9.1	0.4	8.5 to 10.2
Stability (10^−7^ m^−1^)	32.8	23.1	−0.2 to 74.7	37.5	20.7	0.7 to 65.9
Surface Nitrate (μM)	2.9	3.9	BDL * to 11.2	2.9	4.1	BDL * to 11.4
Bottom Nitrate (μM)	9.4	5.4	0.4 to 15.9	12.2	3.2	5.8 to 15.7
Surface Phosphate (μM)	0.9	0.2	0.5 to 1.6	1.0	0.2	0.5 to 1.5
Bottom Phosphate (μM)	1.6	0.5	1.0 to 2.4	1.7	0.6	0.8 to 3.3
Surface Silicic Acid (μM)	2.1	1.2	1.1 to 6.5	1.3	1.3	0.2 to 4.8
Bottom Silicic Acid (μM)	4.4	2.9	0.6 to 9.3	5.8	2.5	0.3 to 9.3
Water column depth (m)	80.2	15.8	39 to 101	83.1	15.9	37 to 107
δ^15^N (‰)	11.6	1.8	5.5 to 13.6	9.1	1.5	6.7 to 11.4
δ^13^C (‰)	−20.2	1.7	−24.8 to −17.3	−21.4	1.3	−24.0 to −18.7

* BDL Below detection limit.

**Table 2 biology-13-00990-t002:** Summaries of the selected parsimonious GLMM models with the best predictors.

	Parameter	SD	DF	*p*-Value
δ^15^N				
Intercept	11.57	0.87	126	<0.0001
Surface nitrate	−0.50	0.08	30	<0.0001
Surface silicate	0.61	0.21	30	0.0085
Stability	−0.24	0.11	30	0.0481
δ^13^C				
Intercept	−14.92	1.33	126	<0.0001
Surface nitrate	−0.19	0.07	30	0.0107
Stability	−0.36	0.14	30	0.0178
Depth	−0.03	0.01	30	0.0384

**Table 3 biology-13-00990-t003:** Summary of observed copepod species and developmental stages, including their mean abundances and standard deviations.

Copepods	Mean Abundance ± S.D. 2016	Mean Abundance ± S.D. 2017
Total copepods	554 ± 641	222 ± 120
Copepodites IV–V	109 ± 107	93 ± 62
Adults	445 ± 556	129 ± 71
*Calanoides carinatus*	6 ± 5	4 ± 2
*Ctenocalanus vanus*	409 ± 538	120 ± 73
*Acartia tonsa*	33 ± 45	12 ± 18
*Drepanopus forcipatus*	39 ± 47	5 ± 4
*Oithona aff. helgolandica*	11 ± 12	3 ± 1
*Oithona atlantica*	5 ± 3	3 ± 2

## Data Availability

The data presented in this study are available on request from the corresponding author. Data disclosure is at the discretion of the author of the communication.

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
