# Peer review of "A Strong Link Between Oceanographic Conditions and Zooplankton δ13C and δ15N Values in the San Jorge Gulf, Argentina"

_biology, 2024, doi:10.3390/biology13120990_

Round 1

Reviewer 1 Report

Comments and Suggestions for Authors

Review for the paper "A strong link between oceanographic conditions and zooplankton δ13C and δ15N values in San Jorge Gulf, Argentina" by Galván DE, Funes M, Paparazzo FE, Alonso Roldán V, Derisio C, Pisoni JP, Temperoni B, del Valle DA, Segura V and Newsome SD submitted to "Biology".

General comment.

Mesozooplankton are essential trophic links in marine food webs, transferring energy and materials by linking the microbial food web to higher trophic levels. Copepods are a diverse assemblage that dominate mesozooplankton communities. The role of copepods in planktonic food webs can be determined by their overall trophic level relative to primary producers. Because copepods are highly dependent on phytoplankton as prey, seasonal and spatial changes in phytoplankton composition and availability affect the abundance and feeding behavior of copepod assemblages. Coastal and estuarine environments often experience rapid fluctuations in inorganic carbon and nitrogen inputs. The δ13C values of suspended particulate organic matter in coastal systems increase from the head to the mouth of a bay, while the δ 15N values of primary producers increase from nutrient-sufficient to nutrient-limiting, and are particularly high with anthropogenic wastewater nitrogen inputs. Because copepod isotope values vary with food source availability, seasonal and spatial patterns generally follow trends in their food sources or dominant prey. This paper examines the spatial and temporal variability of stable isotope values in a gulf in Argentina. The authors showed that environmental factors shaped the composition of two major stable isotopes in medium-sized copepods during spring periods. The research provides a valuable contribution to the current knowledge of zooplankton ecology in coastal ecosystems and may be of interest to scientists concerned with biogeochemical cycles in shelf and coastal waters. In general, the paper is well written and structured. The methods are appropriate and correctly applied. The discussion gives a relevant interpretation of the main results. However, the main results are not well visualized. In particular, some important figures are missing in the MS, which makes it difficult to assess the presentation of the main data.  The quality of the paper is high.

Specific remarks.

Introduction. To help the reader put the results of this study into context, it would be helpful to outline some general aspects of zooplankton composition, occurrence, distribution and abundance in the Argentine Sea.

Materials and methods. It is suggested to include a table showing the location of sampling stations, sampling dates, depth and ID. This may be done by including an additional table.

Materials and Methods. Salinity has been identified as one of the important drivers of zooplankton/copepod assemblages in coastal regions. Explain why salinity was not included in the analysis.

Results. A table showing the composition and abundance of particular copepod taxa must be included in the results. It would also be good to show the contribution of copepod taxa (abundance) to the total zooplankton.

Reviewer 2 Report

Comments and Suggestions for Authors

Understanding how δ13C and δ15N values in San Jorge Gulf, Argentina.It has reference significance for Marine ecological environment protection.The article is also rich in pictures and tables,totally,the manuscript is well-written. However, I have a few minor revisions that should be addressed to improve the manuscript: 

line 24 "a proper"should Leave one space empty

line27 could introduce baseline isoscapes first

lin 118-153 is too long,Please refine your language

line163Add research significance

line179why samples stations not same?"Zooplankton samples were collected at 21 stations in 2016 and 14 stations in 2017",Whether different sampling points affect the description of spatiotemporal changes?

line 238 What is the "QQplot"mean?

line275-280 describe the difference between two years in more detail

Material and Methods:Why just choose generalized linear mixed models (GLMM), is there a better model to analyze? or multiple models for comprehensive analysis?

Summary analysis is missing in the results and discussion sections. It is recommended to add a summary statement so that people can understand the overall results.

Please make the discussion more logical.

Comments on the Quality of English Language

The quality of the English language can be improved.
